# Overview of and First Observations from the TILDAE High-Altitude Balloon Mission

**Bennett A. Maruca**[1,2]**, Raffaele Marino**[3,2]**, David Sundkvist**[2]**, Niharika H. Godbole**[2]**, Stephane Constantin**[4]**, Vincenzo Carbone**[5]**, and Herb Zimmerman**[6]

[1]Department of Physics and Astronomy, University of Delaware — Newark, DE 19716, USA
[2]Space Sciences Laboratory, University of California, Berkeley — Berkeley, CA 94720, USA
[3]Laboratoire de Mécanique des Fluides et d'Acoustique, CNRS, École Centrale de Lyon, Université de Lyon — Ecully, France
[4]Modular Robotics — Boulder, CO 80301, USA
[5]Dipartimento di Fisica, Università della Calabria — 87036 Arcavacata di Rende, Cs, Italy
[6]Applied Technologies — Longmont, CO 80501, USA

*Correspondence to:* B. A. Maruca (bmaruca@udel.edu)

**Abstract.**

Though the presence of intermittent turbulence in the stratosphere has been well established, much remains unknown about it. In-situ observations of this phenomenon, which have provided the greatest detail of it, have mostly been achieved via sounding balloons (i.e., small balloons which burst at peak altitude) carrying constant-temperature "hot wire" anemometers (CTA's). The Turbulence and Intermittency Long-Duration Atmospheric Experiment (TILDAE) was developed to test a new paradigm for stratospheric observations. Rather than flying on a sounding balloon, TILDAE was incorporated as an "add-on" experiment to the payload of a NASA long-duration balloon mission that launched in January, 2016 from McMurdo Station, Antarctica. Furthermore, TILDAE's key instrument was a sonic anemometer, which (relative to a CTA) provides better-calibrated measurements of wind velocity and a more-robust separation of velocity components. During the balloon's ascent, TILDAE's sonic anemometer provided atmospheric measurements up to an altitude of about 18 km, beyond which the ambient air pressure was too low for the instrument to function properly. Efforts are currently underway to scientifically analyze these observations of small-scale fluctuations in the troposphere, tropopause, and stratosphere and to develop strategies for increasing the maximum operating altitude of the sonic anemometer.

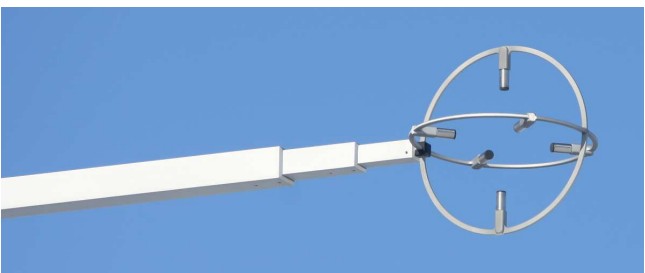

**Figure 1.** Photograph of the sonic anemometer used for TILDAE. The sensor head (right) incorporates six transducers, which are mounted in the cylindrical housings that protrude inward from the metal rings. The effective distance between oppositely positioned transducers is approximately 10 cm.

## 1 Introduction

The Turbulence and Intermittency Long-Duration Atmospheric Experiment (TILDAE or TILDÆ; PI's R. Marino and B. A. Maruca) was developed to make high-cadence, in-situ measurements of the stratosphere's velocity fields. The central instrument of TILDAE was a sonic anemometer, which is shown in Figure 1. As detailed below, this type of device measures the transit time of ultrasonic pulses through the air. This techniques provides well-calibrated measurements not only of the wind velocity but also of the air's sound speed (and thus temperature).

One of the primary goals of TILDAE was to characterize the energy transfer in a stratified atmosphere that results from the interplay of waves and turbulence (Bartello, 1995; Smith and Waleffe, 2002; Marino et al., 2015a). Gaining knowledge on the spectral distribution of energy in the stratosphere is indeed of major importance for improving parametrization schemes and spatial resolution in climate models (Skamarock et al., 2014) and thus for increasing their predictive power. From a fundamental point of view, insights from TILDAE stand to help verify theoretical predictions of statistical mechanics concerning the condensation of quadratic invariants in anisotropic turbulence (Kraichnan, 1975; Herbert et al., 2014) and to validate solutions from high resolution direct numerical simulations in the Boussinesq framework (Marino et al., 2015b). Finally measurements of the three velocity-components and temperature at a high temporal resolutions would allow for a global assessment of the properties of intermittency in the upper atmosphere.

TILDAE was developed not only as a scientific experiment but also as an engineering study of operating a commercial sonic anemometer in the low temperatures and pressures of the stratosphere. Though sonic anemometers have been employed for microscale meteorology since the seminal work of Suomi (1957), their use has been largely limited to the troposphere. Even so, Ovarlez et al. (1978) did develop a specialized sonic anemometer that was incorporated by de la Torre et al. (1994, 1996) into the payload of a high-altitude balloon and used to explore stratospheric waves associated with the Andes Mountains. Sonic anemometers have also been flown at lower altitudes on airplanes (Cruette et al., 2000) and tethered balloons (Canut et al., 2016).

TILDAE recently flew as an "add-on" experiment to GRIPS (Gamma-Ray Imager/Polarimeter for Solar flares) (Shih et al., 2012; Duncan et al., 2016), which was launched on a long-duration zero-pressure balloon that typically flies for 7 to 15 days and at altitudes up to about 40 km (NASA, 2015). After it's initial ascent, this type of balloon travels at nearly the velocity of the wind[1] and changes altitude very smoothly and gradually (largely in response to diurnal variations in solar elevation and the slow loss of helium from the balloon). Thus, TILDAE spent extended periods of time in interface regions between stratification layers and was poised to make observations of the development and dissipation of turbulence therein.

During the ascent of the GRIPS balloon, TILDAE successfully provided measurements up to an altitude of about 18 km. Above this altitude, TILDAE's electronics continued to function, but, as detailed below, the anemometer re-

turned only preset "fill values." Though it had been hoped that measurements could be collected during the balloon's "float" phase, this outcome was not entirely unexpected. The extreme temperatures and pressures pushed the limits of the sonic anemometer, which had received only relatively modest adjustments to its "off the shelf" configuration. Nevertheless, TILDAE's anemometer did function to a high enough altitude to provide scientifically useful measurements from the tropopause and stratosphere.

TILDAE did not rely on any particularly unique feature of the GRIPS gondola as it was simply taking advantage of the flight opportunity that GRIPS provided. Even so, the behavior of the sonic anemometer during this flight has provided new insights into how its performance could be optimized for use around the tropopause and in the lower stratosphere. The results from TILDAE motivate future exploration of these regions with sonic devices on dedicated missions.

This article presents an overview of TILDAE, the events of its flight, and a preliminary scientific analysis of data therefrom. Full scientific analysis has been reserved for one or more future publications. Section 2 of this article introduces the fundamental principles of sonic anemometers in general. The TILDAE system (i.e., its sonic anemometer and support electronics) are detailed in Section 3. Section 4 describes TILDAE's performance during flight both from a technical and scientific standpoint. The interpretation of these results is then discussed in Section 5. Preliminary conclusions about the TILDAE mission are presented in Section 6.

## 2 Sonic Anemometers

The central instrument of TILDAE was a sonic anemometer, which measured both wind velocity and sound speed. While a full review of sonic anemometers is beyond the scope of this article, this section is included to provide basic information on these devices (Section 2.1) and to compare them to hot-wire anemometers (Section 2.2), which have been more widely used in observations of the stratosphere. For additional information, interested readers may consult Kaimal (1978) and Wyngaard (1981), who review the early development of sonic anemometers, and Cuerva and Sanz-Andrés (2000), who detail the theory of sonic-anemometer design and operation.

### 2.1 Principals of Operation

Credit for the first, practical sonic-anemometer is usually given to G. F. Carrier & F. D. Carlson, who worked at Harvard University's Cruft Laboratory[2] under a 1944 contract from the United States Office of Scientific Research and De-

---

[1]The balloon is carried by the wind at the balloon's altitude, but the gondola hangs a significant distance below that. Therefore, wind-velocity measurements from the gondola are effectively of the vertical sheer of the wind (see, e.g., Section 2.1 of Theuerkauf et al., 2011).

[2]Various sources, (at least as far back as Whelpdale, 1967) have misidentified the name of this facility as either "Craft" or "Croft."

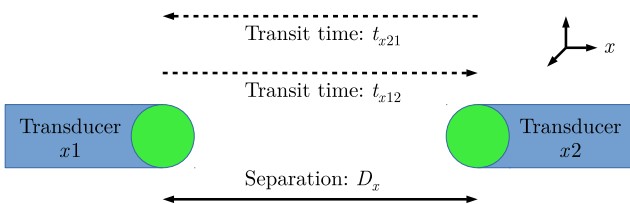

**Figure 2.** Diagram of a pair of a transducers from a sonic anemometer.

velopment (OSRD)[3] to develop an anemometer for use in measuring aircraft speed.[4] Though various refinements have been introduced (e.g., measurements along multiple axes), the basic principles of a sonic anemometer's design and operation have remained largely unchanged.

Figure 1 shows a photograph of TILDAE's sonic anemometer (see Section 3.1 for the technical details of this specific instrument). This anemometer has six, piezoelectric transducers that are arranged around the sensor head and directed inward. Each transducer forms a pair with the transducer opposite it and is used to send ultrasonic "chirps" to and to receive such signals from its mate.

Figure 2 shows a diagram of a pair of transducers from a sonic anemometer. The $x$-axis has been arbitrarily chosen to align with this pair, which are separated by a distance $D_x$. The propagation time of a sound wave from Transducer $x1$ to Transducer $x2$ is denoted as $t_{x12}$, and that of a sound wave from Transducer $x2$ to Transducer $x1$ as $t_{x21}$. Thus, if $D_x$ is known, and $t_{x12}$ and $t_{x21}$ are measured, then the $x$-component of the wind speed can be inferred to be

$$v_x = \frac{D_x}{2}\left(\frac{1}{t_{x12}} - \frac{1}{t_{x21}}\right) . \tag{1}$$

Likewise, the speed of sound in the air is

$$c = \frac{D_x}{2}\left(\frac{1}{t_{x12}} + \frac{1}{t_{x21}}\right) , \tag{2}$$

which corresponds to an air temperature of

$$T = (273.15 \text{ K})\left(\frac{c}{331.47 \text{ m s}^{-1}}\right)^2 . \tag{3}$$

The denominator of the fraction in expression above is the speed of sound in dry air at $T = 273.15 \text{ K} = 0°$ C as reported by Smith and Harlow (1963).

---

[3]Kaimal and Businger (1963), Whelpdale (1967), and various authors since have erroneously attributed Carrier & Carlson's work on the sonic anemometer to the National Defense Research Committee (NDRC), which, in 1941, was superseded as a funding agency by the newly formed OSRD.

[4]OSRD Division 17, Contract 658 (July 1, 1944). Though the results of this work do not seem to have ever been published in any peer-reviewed journal (probably due to the wartime classification of OSRD activities), notes and other records from this project are currently publicly archived by the Technical Reports and Standards (TRS) Unit of the United States Library of Congress.

In greater generality, Equation 3 would have a correction for the relative humidity of the air. Nevertheless, in cold air (such as that of the upper troposphere and lower stratosphere), the vapor pressure of water is so low that the humidity correction has limited effect. Even at $T = 0°$ C, the difference in sound speed between dry air and saturated air is less than $1 \text{ m s}^{-1}$ (Cramer, 1993).

## 2.2 Comparison to Hot-Wire Anemometers

Another type of anemometer is the "hot-wire" anemometer, which is widely used in atmospheric sciences and other fields of research to measure the flows of fluids (both gases and liquids). As such, some comparison of sonic and hot-wire anemometers is warranted.

A hot-wire anemometer consists of a short length of fine wire, which is heated by allowing an electrical current to pass through it. The resistivity of the wire's metal varies with temperature. A constant-temperature anemometer (CTA) is a type of hot-wire anemometer in which a feedback circuit adjusts the amount of current passing through the wire to maintain a constant resistance and thus a constant wire-temperature (Whelpdale, 1967; Rathakrishnan, 2007). Therefore, a measurement of the power expended in heating the wire indicates the cooling rate of the wire, which is highly dependent on the velocity (speed and direction) the wind blowing on it.

Perhaps the greatest advantage of hot-wire anemometers over their sonic counterparts is their compact size. Indeed, in some cases, the wire is less than one centimeter long. This makes hot-wire anemometers highly suitable for inclusion in low-mass payloads (such as those for hand-launch balloons). Such small sizes also help to provide faster response times, which (in turbulence studies) allow the sampling of smaller scales (see, e.g., Gerding et al., 2009; Theuerkauf et al., 2011). Furthermore, unlike some other types of anemometers, hot-wires are most sensitive to low rates of flow and are therefore particularly useful for measuring, e.g., low-speed wind.

Nevertheless, hot-wire anemometers carry significant limitations. In particular, calibration of these instruments can be extremely challenging because the cooling rate of the wire depends not only on the wind velocity but on the air's density, temperature, and composition as well (Wyngaard, 1981; Rathakrishnan, 2007). The highly non-linear nature of these dependencies make absolute calibration difficult without reproducing control conditions in a laboratory. In contrast, sonic anemometers have far more linear responses that are readily converted to an absolute scale (Kaimal, 1978). Though sonic anemometers do have their own complications (e.g., transducer shadows), these can often be compensated for as part of a robust calibration scheme (Wyngaard and Zhang, 1985).

A tremendous advantage of sonic anemometers is their ability to measure air temperature (which, in contrast, is

a liability with hot-wire anemometers). Though thermistors and thermocouples can be used for this purpose, they suffer from slow response-times (especially at low air-pressures) and are prone to spurious heat-flow from incident sunlight and from their housings and supporting structures. Conversely, sonic anemometers can provide high-frequency air-temperature measurements that are far less contaminated by these effects and that are synchronized with the measurements of wind velocity (Barrett and Suomi, 1949). This allows them to be used to detect even small-scale fluctuations in temperature and to explore how they correlate with fluctuations in the velocity field (see, e.g., Larsen et al., 1993).

A notable limitation of the sonic anemometers is their difficulty in making accurate measurements during fog or precipitation, when frost or dew have accumulated on them, or when the air contains appreciable smoke or dust. Though the instruments themselves can be very robust, liquid droplets and solid particles in the air can deflect the sound waves emitted by the transducers and corrupt the measurements (Kaimal, 1978).

## 3 TILDAE System

TILDAE essentially consisted of two subsystems: the sonic anemometer, which measured wind velocity and sound speed, and the electronics box, which provided power and processed measurements. These are discussed in Sections 3.1 and 3.2, respectively.

This Section contains considerable detail on TILDAE's engineering. These details are included here in the hopes that they might be useful to future teams developing small payloads for high-altitude balloons.

### 3.1 Sonic Anemometer

TILDAE's sonic anemometer, which is shown in Figure 1, was a "V-Style" probe from Applied Technologies with customized configuration and calibration to improve performance in low pressure environments. As with the other models produced by Applied Technologies, this anemometer featured a rugged construction from anodized aluminum, a wide range of operating temperatures, a moderate power consumption ($\lesssim 1$ W), and a configurable RS-232 serial-interface. This particular model was selected for TILDAE for two reasons. First, its transducer-pair axes are fully orthogonal, which allows for the most accurate separation of the components of wind velocity. Second, the V-Style has the shortest spacing between transducers (10 cm), which maximizes the fidelity of the audio signals passing between paired transducers.

The decrease in the anemometer's sound quality with air-pressure was a major concern for the development of TILDAE. To improve the low-pressure performance of TILDAE's sonic anemometer, the gain levels on its audio

amplifiers were adjusted based on tests of the instrument at various pressures in a vacuum chamber. The ultimate gain-levels were selected conservatively as too much gain can induce noise or even crack a transducer's piezoelectric crystal. The tests indicated that the modified anemometer would operate at altitudes well above the Antarctic tropopause, but exact performance-predictions were difficult because a thermal vacuum chamber (which would have simulated not only air pressure but also air temperature and solar illumination) was unavailable.

The sonic anemometer measured wind velocity, $\boldsymbol{v} = (v_x, v_y, v_z)$, and sound speed, $c$, with a precision of $0.01 \text{ m s}^{-1}$ and at a cadence of 200 Hz. The measurements of $c$ were derived from the same pair of transducers used to measure $v_z$.[5] All measurements were sent via the serial interface to the TILDAE electronics box for data storage.

The sonic anemometer was mounted to the GRIPS gondola via a support arm that was constructed from aluminum square-tube and that protruded horizontally from the port (i.e., left) side of the gondola (for a diagram of the GRIPS gondola that shows TILDAE's placement thereon, see Figure 3 of Duncan et al., 2016). The arm was affixed to the gondola just below its azimuth motor and extended the anemometer's sensor head approximately 1.7 m from that point. The anemometer was oriented so that its $z$-axis was vertically aligned. The gondola was ultimately flown on a "40 MCF" ($39.57 \times 10^6 \text{ ft}^3 = 1.12 \times 10^6 \text{ m}^3$) balloon (for the typical geometry and flight line, see NASA, 2015).

GRIPS, being a solar observatory, carried a fully-automated pointing-control system (Shih et al., 2012; Duncan et al., 2016). Two motors controlled the orientation of the gondola: one rotated the entire gondola to establish its azimuth, and the other rotated the telescope boom to the appropriate elevation. TILDAE's anemometer was affixed to GRIPS' gondola itself (not its boom), and thus benefited from GRIPS' azimuth-tracking of the Sun. So long as the pointing-control system was active, the gondola's position and time (as recorded from GRIPS' on-board GPS receivers) could be used to rotate the anemometer's $x$- and $y$-axes into an Earth-based coordinate system.

Since GRIPS' pointing-control system would orient the gondola relative to the Sun (versus the ambient flow of air), there would inevitably be periods during which the measurements of TILDAE's anemometer would be contaminated by the gondola's wake. Nevertheless, the pointing-control system's slow and gradual rotation of the gondola (i.e., about one rotation per day) would mean that there would be periods during which the anemometer could make unobstructed measurements of the wind. On-board sensors (e.g., GRIPS' dif-

---

[5]In principle, any of the three axes could used to derive measurements of $c$: theoretically, they should all return the same value. The choice of the $z$-axis for the TILDAE anemometer was an entirely arbitrary one.

ferential GPS system and TILDAE's accelerometer) would assist in identifying these periods.

Unfortunately, as detailed in Sections 4 and 5, delays in commissioning and calibrating the pointing-control system meant that it was not fully active (i.e., properly tracking the Sun) until the the gondola was at float altitude. Additionally, it is speculated that electrical or vibrational interference from the pointing motors may have resulted in narrow-band noise that has been identified in some portions of the TILDAE flight data.

## 3.2 Electronics Box

The TILDAE electronics box provided power to and processed data from the sonic anemometer. It also served as TILDAE's electrical and electronic interface with GRIPS. Figure 3 shows a simplified system-diagram of the box that includes its internal components as well as the external components to which it connected.

GRIPS provided power for the TILDAE system in the form of an unregulated DC current at a nominal voltage of $+28$ V. The TILDAE box's power supply converted this into two, regulated levels: $+12$ V to power the anemometer and $+7$ V to power the remaining electronics in the box.

The TILDAE box also contained an Arduino Due microcontroller board, which processed and stored all data. The Due features a reasonably high clock-speed, moderate power requirements, and numerous digital and analog input/output lines. Like many microcontroller boards in the Arduino family, the Due supports stackable expansion-boards known as "shields." TILDAE incorporated two such shields: the standard Arduino Ethernet Shield, and a custom-designed TILDAE shield.

The Ethernet shield provided the microcontroller with an Ethernet interface, through which it connected to GRIPS' flight network. The microcontroller utilized this connection to provide the GRIPS flight computer with "housekeeping" packets, which included data on the health of the TILDAE system and averaged-down measurements from the sonic anemometer. These packets were enqueued for transmission via GRIPS' telemetry streams. The full-cadence measurements from the sonic anemometer (which could not be telemetered due to limited bandwidth) were stored on a microSD card via an built-in interface on the Ethernet shield.

The TILDAE shield was designed to serve various functions. Like the commercial Ethernet shield, it incorporated a microSD card interface so that a fully redundant copy of all measurements could be stored. The TILDAE shield also included the RS-232 interface that enabled the microcontroller to receive measurements from the sonic anemometer. Two types of housekeeping sensors were likewise supported by the TILDAE shield. First, an on-board, MEMS accelerometer was used to characterize the vibrations of the GRIPS gondola. Second, a set of solid-state temperature sensors extended from the TILDAE shield to monitor the temperatures of various components in the box.

The electronics box itself, a photograph of which is shown in Figure 4, was custom-built from aluminum. The power supply was placed in a partitioned portion of the box (with RF-filtering feedthroughs for power in and out) to help isolate noise from the DC/DC converters. This design was developed to avoid potentially interfering with some of GRIPS' sensitive, analog electronics.

Special care was also given to ensure that the electronics box could operate in the stratosphere. The low atmospheric-pressure meant that (despite the low air-temperature), heat could easily build up in electronic components. At-risk components were identified via thermal imaging (see, e.g., Figure 5). Custom, copper heat sinks were fabricated and installed to increase the surface area for thermal radiation and to provide a path to conduct heat from these components to the box itself. The effectiveness of this thermal management was confirmed prior to flight via testing in a vacuum chamber.

The level of background radiation also significantly increases with altitude and posed a risk for data storage. Consequently, the microSD cards chosen for TILDAE utilized single-level cell (SLC) versus multilevel cell (MLC) technology (Sanvido et al., 2008). Each TILDAE data-packet (either stored on the cards or telemetered) included a "checksum" value for validation.

## 4 Results

The flight of TILDAE on-board GRIPS is described in Section 4.1. A major goal for TILDAE was to establish the maximum operating altitude of the sonic anemometer and to develop ways that it could be increased further. As detailed below, the anemometer returned valid measurements up to an altitude of 18 km. While it had been hoped that TILDAE would collect measurements during the balloon's "float" phase (at altitudes above 30 km), the ascent-phase measurements have provided scientifically useful observations of the tropopause and the lower stratosphere.

In Section 4.2, TILDAE's temperature measurements are compared to those from meteorological radiosondes that were launched on the same day and from a nearby site. Early results from the ongoing scientific analysis of these data are reported in Section 4.3.

## 4.1 Launch and Flight

GRIPS was launched at approximately 01:41 UTC on January 19, 2016 from the Long-Duration Balloon (LDB) facility at McMurdo Station, Antarctica ($77.85°$ S, $167.22°$ E). The flight lasted 11 days, 19 hours, and 50 minutes, during which time Antarctica's polar vortex carried the balloon more than half way around the continent. TILDAE (including its two microSD-cards with full-rate anemometer-

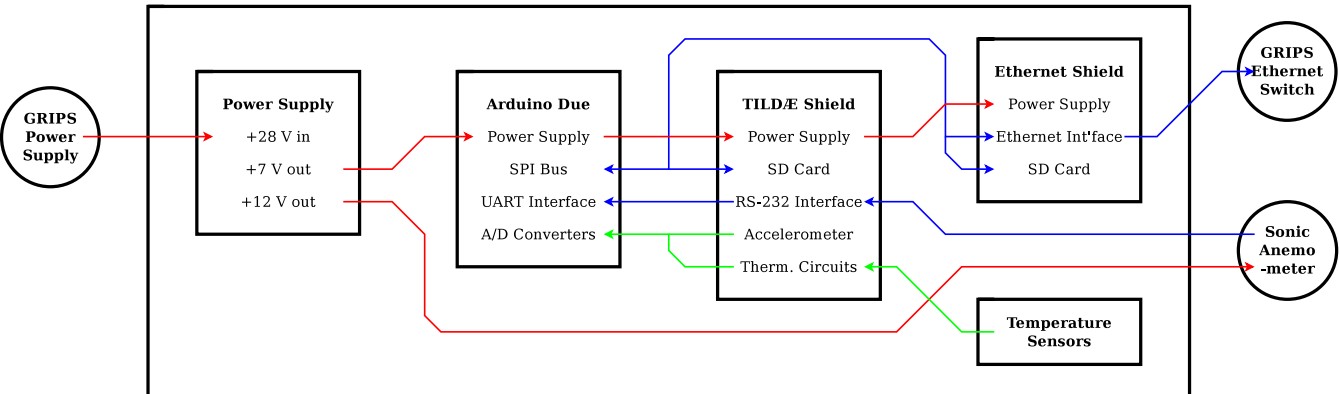

**Figure 3.** Simplified system-diagram of TILDAE's electronics box. The key components within the box are shown in rectangles while external components are shown in circles. Arrows indicate the connections between components and are colored based on their function: red for power, blue for digital signals, and green for analog signals.

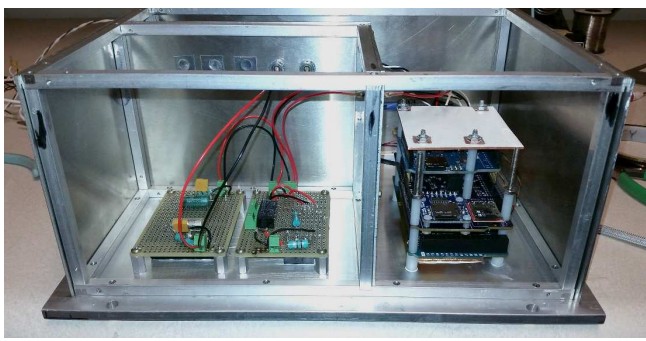

**Figure 4.** Photograph of TILDAE's electronics box with some of its panels removed. The power supply is visible on the left, and the microcontroller "stack" on the right.

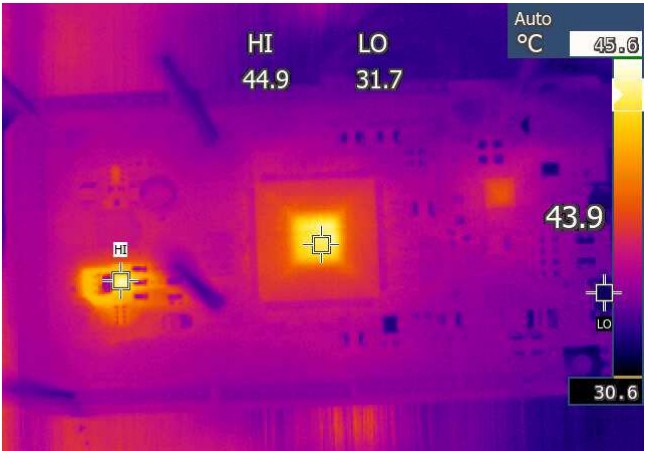

**Figure 5.** Thermal image of the Arduino Due during operation.

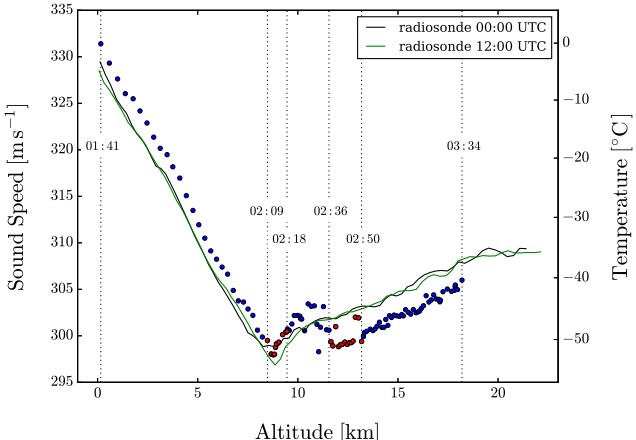

**Figure 6.** Plot of sounds speed versus altitude as measured by TILDAE's sonic anemometer. The right-hand axis equivalent air temperature (per Equation 3). The data shown in the plot have been averaged down to a one-minute cadence. The red data points (i.e., those between 02:09 and 02:18 UTC and between 02:36 and 02:50 UTC) are suspect because, as described in Section 4.1, they may suffer from noise contamination. The solid lines show one-minute medians of measurements from two radiosondes (see Section 4.2) that the AMRC launched on the same day as TILDAE at approximately 00:00 (black) and 12:00 (green).

measurements) was successfully recovered from the landing site several days later.

Figure 6 shows a plot of the sound speed (measured by TILDAE's sonic anemometer) as a function of the gondola's altitude (measured by GRIPS' on-board GPS-system). Each point on the plot represents a one-minute median.

Based on the TILDAE data in Figure 6, the gondola crossed the tropopause at approximately 02:10 UTC and at an altitude of approximately 8.0 km, which is consistent with typical values for this location and time of year (Hoinka,

1999). Prior to this point, the air temperature (as inferred from the sound speed) dropped steadily with altitude; afterward, the temperature leveled off and then began to rise. Notably, the balloon's ascent profile also changed significantly at about this altitude. Immediately below it, the balloon was ascending relatively steadily at an average rate of $4.6\,\mathrm{m\,s^{-1}}$; above it, the ascent became more irregular and the average rate dropped to $2.0\,\mathrm{m\,s^{-1}}$.

Occasionally, the TILDAE anemometer returned non-physical values from one or more of its transducer pairs (e.g., $v \gtrsim 50\,\mathrm{m\,s^{-1}}$ and/or $c \gtrsim 340\,\mathrm{m\,s^{-1}}$). Generally, these could easily be identified in and excised from the time series since they appeared as large, isolated spikes that occurred independently among the transducer pairs. During ground tests, such spikes were very uncommon, and often hours would pass without any. The spikes did become more common immediately after launch, which suggests they may have been induced by vibrations. Even so, during most of the tropospheric portion of the ascent, spikes accounted for only about $0.15\%$ of data from each transducer pair. Furthermore, through most of the lower stratosphere, the spike rate was even lower.

Nevertheless, between 02:09 and 02:18 UTC and between 02:36 and 02:50 UTC (from $8.5$ to $9.5$ km and from $11.6$ to $13.2$ km in altitude, respectively), a large fraction of the measurements returned by the TILDAE anemometer had non-physical values. The spikes during portions of these periods become so common that they came to dominate the data. For this reason, the points in Figure 6 that correspond to these periods have been highlighted and should be treated as suspect. These points are some of the coldest in that plot; while this may simply be noise contamination, it could also indicate genuinely low air temperatures that caused the transducers (which the manufacture has rated only down to $-50°\,$C) to malfunction. The presence of clouds and/or the accumulation of frost on the anemometer's transducers could also have caused these outbursts of noise. Similar measurement-quality issues had been encountered several weeks prior to launch when, during an outdoor test of the anemometer, heavy fog rolled in and deposited frost on the anemometer. Both during the test and the flight, the $z$-axis transducers (which were used to sense both $v_z$ and $c$) seemed more affected by this problem (possibly because, having horizontal surfaces, they might accumulate frost more easily). GRIPS' launch did occur on an overcast day, though details on the height of the clouds were not available.

At 03:22 UTC (16.8 km), the anemometer began to occasionally return a preset "fill value," which indicated that, though the anemometer's digital electronics were functioning, they could not process the signals from the transducers. Almost assuredly, this was the result of the decrease in atmospheric pressure during the ascent. The $z$-axis transducer-pair was the first to exhibit this issue, but all three transducer-pairs were soon affected. Though the occurrence of these fill values was initially infrequent and intermittent, they became more common at about 03:34 UTC (18.3 km). By

03:47 UTC (20.1 km), fill values dominated the data, and, after 04:02 UTC (22.4 km), the anemometer returned nothing but fill values for the sound speed and each component of wind velocity. Though the anemometer's electronics continued functioning for the remainder of the flight, only fill values were ever returned.

## 4.2 Comparison with Radiosonde Measurements

In addition to TILDAE measurements, Figure 6 also shows data from two radiosondes that were launched on the same day (circa 00:00 and 12:00) by the Antarctic Meteorological Research Center (AMRC)[6]. One-minute medians of the radiosonde data were used in the plot.

Overall, measurements from TILDAE's sonic anemometer agree well with those from the radiosondes' temperature probes. In particular, the all three show the tropopause occurring at nearly the same altitude.

Nevertheless, data from the anemometer and the radiosondes are systematically offset from each other by an amount that varies with altitude. Throughout the troposphere, the sonic anemometer's temperatures were several degrees higher, though this difference seems to have slightly decrease with altitude. Then, above an altitude of 13 km, the radiosondes returned higher temperatures than the anemometer, but this difference also decreased with altitude.

These systematic offsets are not entirely surprising. A sonic anemometer uses sound waves to measure the "sonic temperature," which is distinct from the temperature measured by traditional temperature probes (e.g., thermocouples and capacitive devices), which rely on the direct transfer of heat. Sunlight significantly affects temperature probes, and efforts to calibrate for this effect in radiosonde temperature-measurements remains an active area of research (Sun et al., 2013; Ho et al., 2016). GRIPS was launched on an overcast day, so the sunlight exposure of the radiosondes' temperature probes would have increased with altitude. Sonic devices are relatively unaffected by sunlight, but can still be subject to calibration errors due to, e.g., high winds (Huwald et al., 2009; Burns et al., 2012, and references therein). It remains unclear, though, to what extent changes in air pressure can cause similar shifts in the calibration of a sonic anemometer.

## 4.3 Preliminary Scientific-Analysis

TILDAE, during its ascent, recorded science-quality measurements in the troposphere and the lower stratosphere (i.e., below an altitude of about 18 km). The scientific analysis of these measurements is ongoing and will be presented in full detail in one or more forthcoming articles. While the focus of this article is on the technical aspects of TILDAE's system and flight, a preview of this analysis is warranted.

Figure 7 shows plots (versus time) of TILDAE data from a 90-second exemplar-period beginning at 03:04:00 UTC (14.7

---

[6]https://amrc.ssec.wisc.edu/

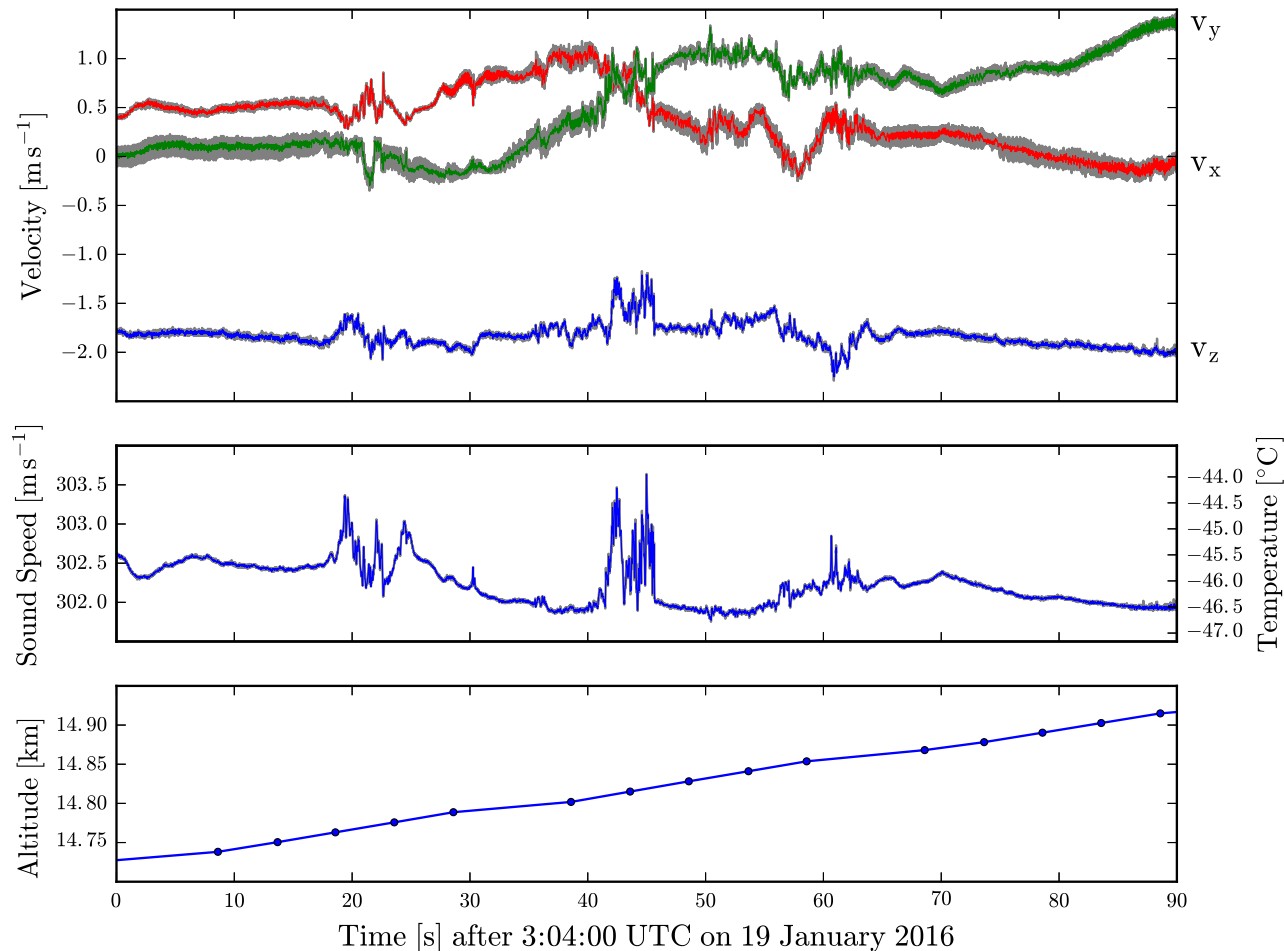

**Figure 7.** From top to bottom, plots (versus time) of wind velocity, air sound-speed/temperature, and gondola altitude. The data in the upper-two plots were measured by TILDAE's sonic anemometer. The raw, 200-Hz measurements are shown in gray, and a three-point median-filtered version is overplotted in color. The $\hat{z}$-axis corresponds to the vertical; thus, the negative $v_z$-values indicate the downward flow of air (in the anemometer's frame of reference) during the balloon's ascent. The bottom plot was generated from GRIPS' GPS data, which were recorded approximately once every 6 or 12 s.

– 14.9 km). The anemometer measurements (upper-two panels) are plotted at their native 200-Hz cadence and are shown both in their raw form and after a three-point median-filter has been applied.

5      At the beginning and the end of this period, the air flow was relatively smooth and steady: the value of each velocity component only changed gradually and on the timescale of seconds. In contrast, the central portion of this period (from about 15 to 70 s after its beginning), the flow is markedly 10 more turbulent. Disruptions are clearly discernible in all three velocity-components and in the sound speed. Furthermore, this enhancement of fluctuations seems to have been concentrated in three, smaller bursts (at about 20, 45, and 60 s).

15      Though the median filter used in Figure 7 is relatively crude, it serves to highlight the intermittent presence of

high-frequency noise in the anemometer measurements. The anemometer's transducer pairs were often affected at different times by this noise. For example, during the period shown in Figure 7, the $z$-axis transducers (i.e., the $v_z$ and $c$ measure- 20 ments) were largely unaffected, but the noise became gradually more pronounced with the $x$-axis transducers and less pronounced with the $y$-axis transducers.

Figure 8 shows the magnification of 6 seconds of wind-velocity measurements from Figure 7. This plot highlights 25 that the transducer noise, when present, generally exhibited an oscillatory pattern, the frequency of which occasionally increased or decreased. Indeed, a preliminary spectral analysis reveals the noise to have been narrowly concentrated at one or two frequencies. 30

Figure 9 shows plots of the power spectral density (for the entire 90-second exemplar-period shown in Figure 7) of three

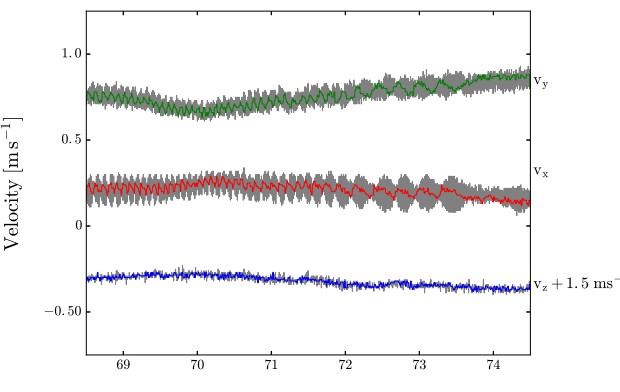

Figure 8. Plot of wind velocity, $v$, for a 6-second portion of the period shown in Figure 7. As in that Figure, the raw, 200-Hz measurements are shown in gray, and the three-point median-filtered data are overplotted in color. A vertical offset has been added to the $z$-component to remove excess whitespace.

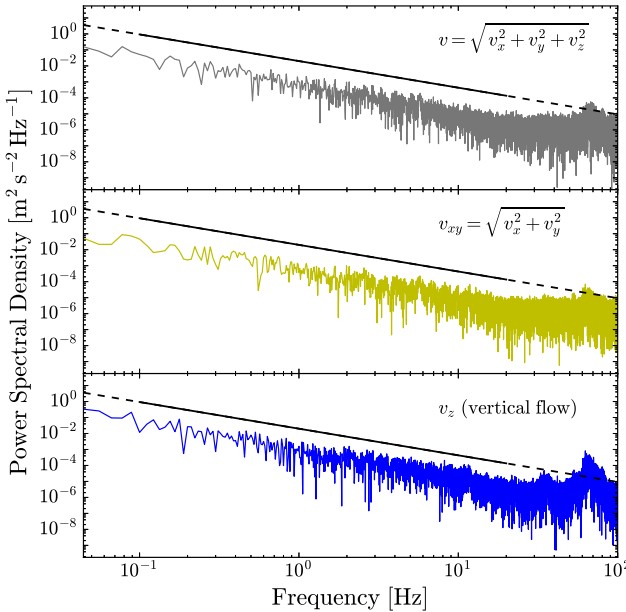

Figure 9. From top to bottom, plots of the power spectral density of the magnitude of velocity ($v$), horizontal velocity ($v_{xy}$), and vertical velocity ($v_z$) for the period shown in Figure 7. No filter was applied to the data used to generate this plot. For reference, a black line is overlaid on each plot and indicates a spectral index of $-5/3$; this line is plotted solid for frequencies over which it agrees well with the data.

quantities: the velocity magnitude ($v$), the horizontal velocity ($v_{xy}$), and the vertical velocity ($v_z$). Specifically,

$$v = \sqrt{v_x^2 + v_y^2 + v_z^2}\,, \tag{4}$$

and

$$v_{xy} = \sqrt{v_x^2 + v_y^2}\,. \tag{5}$$

The power spectral densities were generated using the raw measurements of wind velocity ($v_x$, $v_y$, $v_z$) versus those to which a three-point median filter had been applied.

For frequencies from 0.1 Hz to about 20 Hz, the spectra in Figure 9 closely follow power laws with spectral indices of approximately $-5/3$ (as indicated by the overlaid, black lines). This suggests the presence of well-developed turbulence according to the theory of Kolmogorov (1941a, b, 1991a, b). Furthermore, the close similarity of the spectra in Figure 9 indicates that the turbulence observed during this period was largely isotropic (at least for this range of frequencies).

Above 20 Hz, the spectra in Figure 9 flatten out and exhibit a noticeable spike at about 65 Hz. A second, shorter spike is present (primarily in the $v_z$-spectrum) at about 35 Hz. These features most likely correspond to the high-frequency noise that is evident in the plots of wind velocity in Figures 7 and 8.

## 5 Discussion

The exemplar period explored in Figures 7 − 9 is just one of several bursts of fluctuations that have been identified in TILDAE's measurements from the stratosphere. Many of these appear to be well-developed turbulence, and efforts are underway to quantify them in terms of strength and isotropy. In particular, the following scientific investigations of the TILDAE dataset are currently underway:

- The dissipation rate of kinetic energy provides an important diagnostic for characterizing any turbulent phenomenon. Using techniques such as those developed by Theuerkauf et al. (2011) for single-point balloon measurements, dissipation rates will be inferred for various periods in the TILDAE dataset.

- Intermittency is considered a signature of well developed turbulence. As stratification increases, flows are expected to become more stable and less intermittent. Nevertheless, various studies have reported the opposite both in field gradients (Petoukhov et al., 2008; Lenschow et al., 2012) and in the PDF's of fields themselves (Rorai et al., 2014). Such effects are observed in shear flows (Pumir, 1996), quantum fluids (Baggaley and Barenghi, 2011), and solar-wind plasma (Marino et al., 2012). In the atmosphere, variations in

density stratification with altitude (especially at the tropopause) likely result in a complex evolution of intermittency properties. TILDAE's high-resolution, in-situ measurements of wind velocity will be used to confirm previous observations in the boundary layer and to explore new paradigms for intermittency in the stratified stratosphere.

– The stratosphere's background density profile establishes a preferential direction, which cause its flow to become anisotropic. Unlike in previous experiments based on hot-wire anemometers, TILDAE, via its three-dimensional sonic anemometer, was able to fully separate velocity components parallel and perpendicular to the gravitational gradient. This will allow a detailed study of anisotropy and spectral properties of turbulence above the tropopause. A comparison with previous observations of a proxy of the magnitude of the velocity field vector (Theuerkauf et al., 2011) will also be possible through the evaluation of the isotropic power spectral density (like that in Figure 9) at different altitudes.

– The measured values of the temperature field, which TILDAE's sonic anemometer sampled at the same rate as the wind velocity, will be used to investigate mixing properties. Insights from TILDAE on how energy is transferred across the spectrum and eventually dissipated at smaller scales will be used as a reference to validate parametrization schemes in current weather and climate models. Further analyses will be conducted to investigate turbulence and anomalous mixing in the upper atmosphere by comparing results obtained with TILDAE with statistics based on direct numerical simulations of stably stratified flows (both with and without rotation) (Marino et al., 2014).

The high levels of stratification that characterize the stratosphere produce strong vertical shearing, which allows for the creation of small-scale fluctuations through a turbulent cascade (Marino et al., 2013, 2014). Moreover, turbulence is known to develop in bursts in stratified flows and this feature appears to be robust in the preliminary analysis of TILDAE's measurements. Indeed, the fluctuations evident in Figure 7 are consistent with bursts of stratospheric turbulence observed by Gavrilov et al. (2005) and Haack et al. (2014). Nevertheless, an accurate processing of the data will be needed to filter out the noise components in the measurements and to quantify the levels of intermittency. A further complication to this interpretation is that GRIPS' pointing-control system had not been fully engaged when these measurements were made. The commissioning (i.e., calibration and configuration) of this system required more time than had been anticipated; Sun-tracking was not fully established until the gondola was at float altitude. Thus, the change in horizontal-flow direction seen in Figure 7 may indicate a rotation of the gondola.

The high-frequency noise that is evident in Figure 7 (and at various times throughout the TILDAE dataset) is not expected to deleteriously impact the analysis of turbulence. As is evident from Figure 9, this noise has a relatively narrow bandwidth and occurs at frequencies for which the spectrum is already saturated. One hypothesis for the cause of this noise is electrical and/or vibrational interference from the motors in GRIPS' pointing-control system. This would account for the noise's narrow bandwidth and the variability in its frequencies. Furthermore, a pattern of ringing up/down that is occasionally seen in the noise (e.g., in Figure 8) could be explained by a motor changing speed.

Concurrent with these scientific investigations, efforts are underway to understand the performance of the sonic anemometer during flight and to identify means of improving it. This work currently involves both the analysis of TILDAE's flight measurements and the physical examination of the the sonic anemometer (which was recovered mostly intact from GRIPS' landing site). Attention thus far has been focused on the anemometer's transducers.

Figure 6 indicates that the periods with highest noise in the anemometer measurements seems to correspond with the lowest observed air temperatures, which at times dropped below the manufacturer-specified minimum operating temperature of $-50°$ C for the transducers (see Section 4.1). A new type of transducer is now being investigated that promises to operate down to $-65°$ C, which would facilitate cleaner observations of the tropopause.

Work is also underway to further increase the gain levels of the transducer circuits. As noted in Section 3.1, relatively conservative values were chosen for the flight because, though greater gain increases the anemometer's maximum operating altitude, it also risks damaging its transducers. Early indications suggest that the new type of transducer may be more robust. Further tests (on the ground) are clearly warranted to better establish how high the gain levels can safely be adjusted.

## 6  Conclusions

TILDAE was developed to utilize a three-dimensional sonic-anemometer to make calibrated, high-speed measurements of the temperature and flow of air in the stratosphere. Its flight as an add-on experiment to the GRIPS high-altitude balloon mission was one of only a few-ever scientific attempts to use a sonic anemometer at such altitudes.

The basic specifications of the TILDAE system (and its sonic anemometer in particular) are summarized in Table 1. The anemometer returned consistently valid measurements until the balloon ascended to an altitude of about 18 km, after which its performance steadily declined (due to the reduced air pressure). Though measurements from higher altitudes had been hoped for, TILDAE did successfully provide observations of small-scale fluctuation in the tropopause and lower

**Table 1.** Specifications of TILDAE system as flown

| Measured quantities | $v_x, v_y, v_z, c$ |
|---|---|
| Measurement sensitivity | $0.01 \, \mathrm{m \, s^{-1}}$ |
| Measurement cadence | 200 Hz |
| Minimum operating temperature | $-50°$ C [a,b] |
| Maximum operating altitude | 18 km [b] |

[a] Manufacturer specification
[b] Conclusion from flight test (see Section 4.1)

stratosphere. The scientific analysis of the TILDAE dataset is ongoing.

TILDAE was launched from one of the polar regions, where the height of the tropopause is notably low. It bears mention, though, that the maximum operating altitude that was established for the sonic anemometer, 18 km, is above the typical height of the tropopause over any region of the Earth at any time of year (Hoinka, 1999). Thus, in principle, the TILDAE sonic anemometer, a commercial device with only relatively modest modifications, would be capable of observing any part of the tropopause (at least in terms of pressure).

The results of TILDAE's flight strongly motivate the continued use of sonic anemometers in the exploration of the tropopause and lower stratosphere. While the sonic anemometer itself requires further refinement and ground testing (to optimize performance at these altitudes), a balloon mission dedicated to sonic anemometry would allow the most dramatic improvements over TILDAE. As noted above, TILDAE flew on GRIPS to take advantage of a flight opportunity. On a dedicated flight, though, the gondola could be ballasted to keep it at altitudes at which the anemometer is now known to operate well. Additionally, multiple anemometers could be flown on the gondola so that the effective flow could be more reliably measured. A pointing-control system could be replaced by sensor systems that provide more-regular, higher-cadence measurements of gondola orientation. Such a mission, built on the heritage of TILDAE, would offer a unique platform from which to explore the three-dimensional turbulence of the tropopause and stratosphere.

*Acknowledgements.* We gratefully acknowledge P. Saint-Hilaire and A. Y. Shih for accommodating TILDAE as an add-on instrument to the GRIPS mission. We extend our thanks to the entire GRIPS team for their help and support in the development of TILDAE. We also thank Ball Aerospace for use of their vacuum-chamber facilities and NASA's Columbia Scientific Balloon Facility (CSBF) for the successful launch of GRIPS.

We extend our gratitude to the Antarctic Meteorological Center (AMRC), whose radiosonde program provided the comparison data for Figure 6. We thank the anonymous referees of this article for proposing our use of these data, our addition of Figure 8 and Table 1, and various improvements to the text.

The preparation of this article made use of the SAO/NASA Astrophysics Data System (ADS). BAM thanks L. Marcus, C. Trew, and the entire staff of the Technical Reports and Standards (TRS) Unit of the United States Library of Congress for their assistance with background research.

BAM's contribution to TILDAE was partially funded by the Charles Hard Townes postdoctoral fellowship. VC and RM were partially supported by the Italian Ministry for Education and Research MIUR PRIN Grant No. 2012P2HRCR on "The active Sun and its effects on Space and Earth climate." VC, RM, and BAM acknowledge support from the Turboplasmas project (Marie Curie FP7 PIRSES-2010-269297). RM acknowledges support from the PRESTIGE program coordinated by Campus France (co-financed under Marie Curie FP7 PCOFUND-GA-2013-609102) and the PALSE program at the University of Lyon (through the PALSE postdoctoral fellowship and the action IMPULSION).

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
