# Peer review of "Overview of and First Observations from the TILDAE High-Altitude Balloon Mission"

_Atmospheric Measurement Techniques, 2016_

## Referee Comment (RC1) · Anonymous Referee #2 · 10 Jan 2017

I appreciate the addition of Section 5, where the measurements are discussed in more detail. However, I am still curious on the decent portion of the TILDAE observations and its similarities with the ascent portion showed in the paper.

---

## Author Comment (AC1) · 11 Jan 2017

Thank you for your continued interest in our manuscript.

By "decent," are you referring to the descent portion of the flight? We unfortunately do not have any TILDAE observations from the balloon gondola's descent. NASA's standard procedure for long-duration balloon missions calls for all scientific equipment to be powered down prior to "cut down" (the severing of the gondola from the balloon) to protect the electronics from the landing. Gondolas of this type typically weigh several tons, so, even with parachutes, their descents are relatively rapid, and their landings are frequently rough.

We did consider leaving TILDAE powered on during the gondola's descent. Throughout the flight, we received an 8-Hz, averaged-down stream of TILDAE measurements Printer-friendly version

(via satellite telemetry), so we were aware that the sonic anemometer was not returning measurements at "float" altitudes. We decided, though, that the potential gain of measurements from the comparatively brief descent did not outweigh the risk to the 200-Hz, full-cadence measurements from the ascent that were stored on TILDAE's SD cards.

---

## Referee Comment (RC2) · Anonymous Referee #1 · 3 Feb 2017

The paper by Maruca et al. describes a sonic anemometer for use on a stratospheric long-duration balloon. 3D high-resolved sonic anemometry has a large potential for examination of atmospheric turbulence. Especially questions of isotropy and intermittency are very interesting and only very few techniques are able to answer these demanding tasks. The authors are strongly encouraged to continue data evaluation and the optimization of the instrument. The paper is well written, and several interesting new and partly historic references are appreciated. I suggest some changes in the structure of the paper, taking into account that only limited data has been achieved during the flight (see below). I understand that the scientific analysis of the data just begun and is outside the scope of this paper. Nevertheless the authors should take additional care for quality assurance as well as for the description of the potential of their method. I recommend a review of these sections, with more details given below.

General comments:

Throughout the whole publication the authors concentrate on the description of an instrument that flies piggyback with a large payload on a floating balloon near 40 km altitude. Unfortunately there was no useful data obtained above 18 km, and only unproven suggestions are made to improve the instruments to cover higher altitudes (lower air densities). A lot of ambitious plans are presented and only few have been achieved. From my point of view this flaws the paper to a large extend. What gets lost in the impression of the reader is that there is a very interesting new technique to measure turbulence up to at least  $\sim$ 20 km, with absolute 3D wind and temperature fluctuations at least in the ascent but potentially also during the floating phase – if the floating altitude is low enough. This is not possible with e.g. CTA measurements or Thorpe analysis.

The paper concentrates on the combination with GRIPS. I have not found any information that TILDAE is in principal limited to GRIPS. Maybe other, smaller payloads are possible that would reduce the risk of wake problems or allow for a double TILDAE, measuring on two sides of the payload.

I suggest revising the structure of the manuscript. The whole instrument should be described without the focus on GRIPS and its floating phase at close to 40 km altitude, where TILDAE failed. Of course, this GRIPS ascent would serve as the only test case, but the achievements could be described more prominent. Floating phase measurements and other carrier instruments (if needed) could be described in kind of an outlook.

The description of first tests of sonic anemometry under stratospheric conditions is still possible with the new focus. But one way or the other, a comprehensive test needs further analysis of the regions where TILDAE was unable to measure (tropopause and > 18 km). A pure reference to the specs sheet (and the -50°C limit) is not sufficient, but should be added by e.g. plots of SNR or other meaningful quantities.

Further effort should be put on validation of absolute flow velocities derived from the sonic anemometer. They can, e.g., be compared with the ascent rate (for  $v_z$ ) and with the vertical shear of horizontal winds (for  $v_x$  and  $v_y$ ). Comparative data can be taken from the GPS altitude and position change of the gondola.

Specific comments:

p. 2, I. 8/9: The combination of 3D wind and temperature is very interesting and scientifically interesting. I understand that a full scientific analysis cannot be done within this technical paper. Nevertheless, while this is a core of the scientific potential of TILDAE, I would expect e.g. the spectra for a single case like in Fig. 8 to be shown for all four components instead of the sum of the three wind velocities (see below).

p. 2, l. 17: The GRIPS references say "imager" instead of "interferometer".

p. 2, I. 19/20: For the reader not being familiar with wind soundings on balloons, you should explain that TILDAE is measuring not the true horizontal wind, but the ambient flow around the instrument. This is essentially the vertical shear of the horizontal wind between the altitude of the payload and some "effective altitude" where the wind pushes the balloon-payload-system.

p. 4, l. 14: Please provide some more information on this humidity correction. Fig.
6 contains a lot of tropospheric data. Furthermore, in the Conclusions (p. 15, l. 2/3)
TILDAE is advertised for soundings in the tropopause region.

p. 4, l. 16-18: I do not know about the "long history" of hot-wire anemometry in scientific ballooning. I only know about CTA with its  $\sim$ 10 y history and irregular flights. Furthermore I am surprised that this technique is part of standard radiosondes. From the best of my knowledge radiosondes track the position of the payload either by radar or GPS to get the horizontal winds. Hot-wire anemometers do only catch the flow around the payload, see above.

p. 5, l. 1-4: I would like to see another important property of CTA mentioned here: Its

largest sensitivity at low windspeeds (cf. Whelpdale 1967) provides an advantage for the typically small flow velocities around the payload.

Section 3.1: Could you please provide some further information about TILDAE and GRIPS? For TILDAE the distance between transducers and the accuracy of the wind and temperature measurements (at different pressures) would be interesting to know. For GRIPS the size of the balloon, the length of the whole flight train, and the size of the payload would be interesting for estimation of wake effects. What is the length of the TILDAE boom? "Extending beyond the base of GRIPS" is i) vague and ii) not the most important property to estimate wake effects.

p. 6, l. 13: Is there a special advantage of using  $v_z$  for the temperature measurement? During ascent  $v_z$  is typically larger than  $v_x$  and  $v_y$ ; during the floating phase  $v_z$  is almost zero.

p. 6, I. 28-30: I am not sure whether the wake problem can be identified from the pointing of the balloon and the GPS information. The balloon-gondola system is floating with a horizontal speed being a weighted average over the cross sections of balloon, gondola, ropes etc. Assuming TILDAE is pointing "forward" it could still be in the wake if the wind vector in forward direction at gondola altitude is larger compared to the "mean" wind. The directional information from TILDAE might not help to get the true wind vectors at gondola altitude because it could be influenced by the wake. A second TILDAE in opposite direction would help if the large GRIPS structure is not affecting both at the same time.

p. 7, l. 11-13: Have you been able to recover the SD cards right after landing? If I understand the GRIPS website correctly, most of the GRIPS instrument has been recovered as late as January 2017.

p. 9, I. 15/Fig. 6: I assume that there is a McMurdo radiosonde for 00 UT. It would be interesting to compare the TILDAE temperature data with this standard method. This could furthermore validate the suspect data points.

p. 10, l. 21/22: Radiosonde data provides also information on humid layers (especially if relative humidity is calculated with respect to ice instead of liquid water), even if the sonde was launched a few hours before TILDAE.

p. 11, I. 3 / Fig. 7: Please provide some information about the altitude.

p. 12, l. 4/5: This intermittent structure is in good agreement with other balloon observations, e.g. Gavrilov, Ann. Geophys., 2005, and Haack et al., JGR, 2014.

p. 12, I. 12 / Fig. 8: While TILDAE aims to examine the isotropy of turbulence; it would be interesting to see the individual wind components (and the temperature) instead of the general wind speed.

p. 12, l. 16-18: The spectrum indeed follows the -5/3 slope very nicely. Could you please calculate energy dissipation rates or some other quantity for the description of turbulence? It would help the reader to classify this layer (and estimate the potential of TILDAE).

p. 14, l. 1: On page 6 I read about the advantages from the GRIPS pointing system, but here I learn that this has not been active for the data shown before. I would like to read this much earlier.

p. 14, I. 4: Is there some information about pointing maneuvers in the GRIPS housekeeping data? Instead of wind direction maybe "flow direction" should be used. Other reasons for the apparent change in flow direction might be a true change in wind direction (vertical shear), or a rotation of the gondola due to inertia or wind (flow) pressure on the gondola. Maybe you could infer the pointing direction from GRIPS housekeeping data and compare with horizontal wind direction. Unfortunately these open questions influence the statement that Fig. 8 describes real atmospheric turbulence.

p. 14, l. 8: Do you see any sign of vibration in the accelerometers?

p. 14, l. 10: I am sorry, but I do not see any pattern at 72 s. Please explain.

p. 14, l. 19-22: In order to catch the GRIPS floating altitude, a further density decrease by a factor of  $\sim$ 20 needs to be compensated. Is there any information from the manufacturer whether this gain can be achieved? Is there an automatic control of the gain or might a gain increase result in saturation in the troposphere?

p. 15, l. 1-2: This is a very nice achievement, but flawed by the intention to measure at GRIPS floating altitude.

Typos:

p. 6, l. 13: double "the"

p. 6, l. 17: "based" should read "base"

p. 8, I. 3: "ensuring" should read "ensure"

p. 13, l. 16: "directions" should read "direction"

---

## Author Comment (AC2) · 25 Feb 2017

Based on the comments of Referee #1, we have decided to make various modifications to our manuscript. The attached ZIP file contains both our itemized response to the comments of Referee #1 and our modified manuscript.

Please also note the supplement to this comment: http://www.atmos-meas-tech-discuss.net/amt-2016-359/amt-2016-359-AC2-supplement.zip

---

## Editor Comment (EC1) · J. L. Chau (Editor) · 5 Mar 2017

Given the reviews received and the responses of the authors. I encourage the authors to formally upload a revised version, including an example of at least a 3-D velocity spectrum, to show qualitative the type of of unique measurements they have obtained. I understand a parallel scientific effort is underway, but I think including a resulting 3D spectrum without discussing the scientific details or technical details on how it was obtained, would help the authors to make their current paper (and future papers) more attractive.

---

## Author Comment (AC4) · 10 Mar 2017

We thank the editor for this suggestion. We have modified the manuscript to include separate power-spectra for the horizontal and vertical components of the velocity field along with the spectrum of the velocity magnitude. A set of spectra such as these provides a more complete picture of how kinetic energy is redistributed among scales in different spatial directions in an anisotropic/stratified medium.

Please also note the supplement to this comment:
http://www.atmos-meas-tech-discuss.net/amt-2016-359/amt-2016-359-AC4-supplement.zip

---

## Editor Comment (EC2) · J. L. Chau (Editor) · 11 Mar 2017

Thanks for including the 3D Spectrum. Please be careful that the final submitted version includes such spectrum. I mentioned that since in the system there are two versions, with only one spectrum and the other one with the 3D, 2D and z spectra.

Looking forward to see the final version.

---

## Author Response (AR1)

**Response to Comments from Referee #1**

*The paper by Maruca et al. describes a sonic anemometer for use on a stratospheric long-duration balloon. 3D high-resolved sonic anemometry has a large potential for examination of atmospheric turbulence. Especially questions of isotropy and intermittency are very interesting and only very few techniques are able to answer these demanding tasks. The authors are strongly encouraged to continue data evaluation and the optimization of the instrument. The paper is well written, and several interesting new and partly historic references are appreciated. I suggest some changes in the structure of the paper, taking into account that only limited data has been achieved during the flight (see below). I understand that the scientific analysis of the data just begun and is outside the scope of this paper. Nevertheless the authors should take additional care for quality assurance as well as for the description of the potential of their method. I recommend a review of these sections, with more details given below.*

We thank the referee for their very careful reading and investigation of our manuscript and for providing such extensive comments. The referee has clearly dedicated a great deal of time in researching our project and preparing detailed feedback.

*General comments:*

*Throughout the whole publication the authors concentrate on the description of an instrument that flies piggyback with a large payload on a floating balloon near 40 km altitude. Unfortunately there was no useful data obtained above 18 km, and only unproven suggestions are made to improve the instruments to cover higher altitudes (lower air densities). A lot of ambitious plans are presented and only few have been achieved. From my point of view this flaws the paper to a large extend. What gets lost in the impression of the reader is that there is a very interesting new technique to measure turbulence up to at least ∼20 km, with absolute 3D wind and temperature fluctuations at least in the ascent but potentially also during the floating phase – if the floating altitude is low enough. This is not possible with e.g. CTA measurements or Thorpe analysis.*

*The paper concentrates on the combination with GRIPS. I have not found any information that TILDAE is in principal limited to GRIPS. Maybe other, smaller payloads are possible that would reduce the risk of wake problems or allow for a double TILDAE, measuring on two sides of the payload.*

*I suggest revising the structure of the manuscript. The whole instrument should be described without the focus on GRIPS and its floating phase at close to 40 km altitude, where TILDAE failed. Of course, this GRIPS ascent would serve as the only test case, but the achievements could be described more prominent. Floating phase measurements and other carrier instruments (if needed) could be described in kind of an outlook.*

We freely admit that we were ambitious when we formulated our goals for TILDAE, as our referee would likely agree. Certainly, our experience with TILDAE has taught us much, and, in retrospect, we should have designed the TILDAE mission differently. We hope that our manuscript will convey some of our "lessons learned" to the scientific community.

We have added text to the manuscript's Introduction and Conclusion to emphasize the scientific potential of sonic anemometers in the troposphere and lower stratosphere. We now also explicitly state that TILDAE principally relied on GRIPS for the flight itself – that a future, TILDAE-like system could fly as a dedicated, independent mission. To that end, we also added some specific ideas for the redesign of the TILDAE system.

*The description of first tests of sonic anemometry under stratospheric conditions is still possible with the new focus. But one way or the other, a comprehensive test needs further analysis of the regions where TILDAE was unable to measure (tropopause and > 18 km). A pure reference to the specs sheet (and the -50º C limit) is not sufficient, but should be added by e.g. plots of SNR or other meaningful quantities.*

*Further effort should be put on validation of absolute flow velocities derived from the sonic anemometer. They can, e.g., be compared with the ascent rate (for v_z) and with the vertical shear of horizontal winds (for v_x and v_y). Comparative data can be taken from the GPS altitude and position change of the gondola.*

Unfortunately, we do not have any raw, flight-measurements from the sonic anemometer's transducers. The transducers' voltages are measured by a microcontroller, which only returns the inferred sound speed and velocity components. We did directly measure the transducers' voltages during our ground testing of the sonic anemometer, but these were only conducted in a room-temperature vacuum chamber.

As we state in the manuscript, ground testing in a thermal vacuum chamber would have provided significantly more insight into the performance of the sonic anemometer, but none was available. Nevertheless, given the outcome of the first TILDAE flight, we now know consider tests of the sonic anemometer in such a chamber to be critically important.

*Specific comments:*

*p. 2, l. 8/9: The combination of 3D wind and temperature is very interesting and scientifically interesting. I understand that a full scientific analysis cannot be done within this technical paper. Nevertheless, while this is a core of the scientific potential of TILDAE, I would expect e.g. the spectra for a single case like in Fig. 8 to be shown for all four components instead of the sum of the three wind velocities (see below).*

We appreciate the referee's eagerness to review a 3-D velocity spectrum and a temperature spectrum from TILDAE data. Nevertheless, we strongly feel that such figures (and their accompanying scientific analysis) should be reserved for a separate article.

As the referee notes, such spectra would be rather novel for this area of research. Therefore, adding them to the manuscript would require us to include significantly more material on turbulence theory and observations to fully motivate and interpret the figures. When we began writing this manuscript, we did actually consider doing just that, but we concluded that ultimately it would distract from the discussion of the instrumentation and minimize the results from these spectra. Thus, we decided to reserve all 3-D-velocity and temperature spectra for an article (currently being prepared by coauthor R. Marino) that would be dedicated to that subject.

*p. 2, l. 17: The GRIPS references say "imager" instead of "interferometer".*

We thank the referee for identifying this error, which we have now corrected.

*p. 2, l. 19/20: For the reader not being familiar with wind soundings on balloons, you should explain that TILDAE is measuring not the true horizontal wind, but the ambient flow around the instrument. This is essentially the vertical shear of the horizontal wind between the altitude of the payload and some "effective altitude" where the wind pushes the balloon-payload-system.*

This is indeed an important caveat, and an explicit statement of it has been added to the manuscript.

*p. 4, l. 14: Please provide some more information on this humidity correction. Fig. 6 contains a lot of tropospheric data. Furthermore, in the Conclusions (p. 15, l. 2/3) TILDAE is advertised for soundings in the tropopause region.*

We admit that our original discussion of the humidity correction was somewhat vague, so we have expanded and clarified it. The vapor pressure of water decreases dramatically with temperature. At temperatures typically encountered in the upper troposphere and the stratosphere, it is so low that the relative humidity has very little impact on sound speed. For example, even at 0° C, the difference in sound speed between dry and saturated air is less than 1 m/s (see Cramer, 1993, *JASA*, **93**, 2510).

*p. 4, l. 16-18: I do not know about the "long history" of hot-wire anemometry in scientific ballooning. I only know about CTA with its ～10 y history and irregular flights. Furthermore I am surprised that this technique is part of standard radiosondes. From the best of my knowledge radiosondes track the position of the payload either by radar or GPS to get the horizontal winds. Hot-wire anemometers do only catch the flow around the payload, see above.*

We thank the referee for identifying this misleading statement and apologize for our error. We have revised it to reflect the far more limited use of CTA's in ballooning.

*p. 5, l. 1-4: I would like to see another important property of CTA mentioned here: Its largest sensitivity at low windspeeds (cf. Whelpdale 1967) provides an advantage for the typically small flow velocities around the payload.*

This is indeed an important point, and we have added it.

*Section 3.1: Could you please provide some further information about TILDAE and GRIPS? For TILDAE the distance between transducers and the accuracy of the wind and temperature measurements (at different pressures) would be interesting to know. For GRIPS the size of the balloon, the length of the whole flight train, and the size of the payload would be interesting for estimation of wake effects. What is the length of the TILDAE boom? "Extending beyond the base of GRIPS" is i) vague and ii) not the most important property to estimate wake effects.*

The transducer spacing on TILDAE's sonic anemometer was 10 cm. While we did mention this in the caption of Figure 1, we neglected to do so in the main text. We thank the referee for finding this oversight, which we have corrected. We have also added some more specific details on TILDAE's mounting on GRIPS (e.g., a description of the boom's length and attachment to the gondola) and a statement that GRIPS was carried by a 40 MCF balloon. The typical dimensions and geometry of this type of balloon is described NASA Publication NP-2015-8-326-WFF (which we cite, with this URL).

*p. 6, l. 13: Is there a special advantage of using v_z for the temperature measurement? During ascent v_z is typically larger than v_x and v_y; during the floating phase v_z is almost zero.*

No, the choice of the z-axis was an entirely arbitrary one on the part of the manufacturer, and we have added a footnote to the manuscript to make this clear. In principle, this measurement of the sound speed (and, thus, temperature) should be independent of that of the z-component of wind velocity. Indeed, Barrett & Suomi (1949, *J. Meteorol.*, **6**, 273) developed an early precursor of modern sonic anemometers that only measured air temperature (not wind velocity).

On a future mission, we would very much like to derive temperature measurements from all three pairs of transducers. While, theoretically, pairs should always return the same temperature, this would allow us to validate that expectation and have an added degree of redundancy.

*p. 6, l. 28-30: I am not sure whether the wake problem can be identified from the pointing of the balloon and the GPS information. The balloon-gondola system is floating with a horizontal speed being a weighted average over the cross sections of balloon, gondola, ropes etc. Assuming TILDAE is pointing "forward" it could still be in the wake if the wind vector in forward direction at gondola altitude is larger compared to the "mean" wind. The directional information from TILDAE might not help to get the true wind vectors at gondola altitude because it could be influenced by the wake. A second TILDAE in opposite direction would help if the large GRIPS structure is not affecting both at the same time.*

We agree with the reviewer that a second anemometer would have been a tremendous improvement on the TILDAE system, and, in fact, we briefly considered having two anemometers. Unfortunately, the opposite side of the gondola hosted GRIPS communications equipment, and no protrusions were permitted (lest they interfere with the communication and/or GPS antennae). On a future mission, though, we would include at least two anemometers. This would ensure that, at all times, at least one anemometer is measuring relatively unobstructed air-flow.

*p. 7, l. 11-13: Have you been able to recover the SD cards right after landing? If I understand the GRIPS website correctly, most of the GRIPS instrument has been recovered as late as January 2017.*

Yes, as stated in the first paragraph of Section 4.1, TILDAE's SD cards were recovered. After GRIPS' descent, there was time for one trip of the recovery team the landing site, during which the "data vaults" for GRIPS and its three add-on experiments (including TILDAE) were extracted and returned to their respective science teams.

*p. 9, l. 15/Fig. 6: I assume that there is a McMurdo radiosonde for 00 UT. It would be interesting to compare the TILDAE temperature data with this standard method. This could furthermore validate the suspect data points.*

*p. 10, l. 21/22: Radiosonde data provides also information on humid layers (especially if relative humidity is calculated with respect to ice instead of liquid water), even if the sonde was launched a few hours before TILDAE.*

We are very grateful to the referee for making this suggestion. Until we read it, we sincerely had no idea that radiosondes were regularly flown from McMurdo Station. We have since reached out to the Antarctic Meteorological Research Center, and have augmented Figure 6 with the measurements from two of their radiosondes: the one immediately before and the one immediately after our flight. We have added text to the manuscript that compares the radiosonde and TILDAE measurements.

*p. 11, l. 3 / Fig. 7: Please provide some information about the altitude.*

We have added the altitude range for this 90-second exemplar period to the main text of the manuscript.

*p. 12, l. 4/5: This intermittent structure is in good agreement with other balloon observations, e.g. Gavrilov, Ann. Geophys., 2005, and Haack et al., JGR, 2014.*

We thank the referee for providing us with these excellent references. We have added them.

*p. 12, l. 12 / Fig. 8: While TILDAE aims to examine the isotropy of turbulence; it would be interesting to see the individual wind components (and the temperature) instead of the general wind speed.*

Our response to the referee's comment on Page 2, Lines 8 – 9 of the manuscript addresses this issue.

*p. 12, l. 16-18: The spectrum indeed follows the -5/3 slope very nicely. Could you please calculate energy dissipation rates or some other quantity for the description of turbulence? It would help the reader to classify this layer (and estimate the potential of TILDAE).*

We thank the referee for this useful suggestion. While a detailed characterization of turbulence in the atmospheric layers observed by TILDAE goes beyond the scope of this manuscript, we believe that the evaluation of kinetic-energy dissipation rates is an important diagnostic and a critical element to our scientific analysis. We have now added this analysis as an explicit bullet point in our list (in the Discussion) of our ongoing scientific investigations and included a citation to the useful technique developed by Theuerkauf et al. (2011, *AMT*, **4**, 55) for single-point balloon measurements.

*p. 14, l. 1: On page 6 I read about the advantages from the GRIPS pointing system, but here I learn that this has not been active for the data shown before. I would like to read this much earlier.*

We agree and have added a statement of this caveat to our initial description of the GRIPS pointing control system.

*p. 14, l. 4: Is there some information about pointing maneuvers in the GRIPS housekeeping data? Instead of wind direction maybe "flow direction" should be used. Other reasons for the apparent change in flow direction might be a true change in wind direction (vertical shear), or a rotation of the gondola due to inertia or wind (flow) pressure on the gondola. Maybe you could infer the pointing direction from GRIPS housekeeping data and compare with horizontal wind direction. Unfortunately these open questions influence the statement that Fig. 8 describes real atmospheric turbulence.*

We agree that the interpretation of the 90-second exemplar period shown in Figures 7 – 9 is ambiguous. Indeed, we chose this period to highlight both the potential and the limitations of the TILDAE mission. In our discussion of these figures, we tried to give a balanced presentation of both interpretations of this period: that the fluctuations were induced by air flowing passed the balloon and/or gondola and that they were "native" to the stratosphere. In keeping with this, we agree that using "flow direction" in lieu of "wind direction" is more appropriate and have made this modification to the manuscript.

We have investigated the GRIPS housekeeping data, but it has provided us with relatively few insights. GRIPS flew with a differential GPS system (separate from its regular GPS system), which was capable of measuring the gondola's orientation. Unfortunately, those measurements were only recorded irregularly and infrequently (i.e., every few minutes). GRIPS also carried a quad-cell "Sun sensor" (for pointing control), but it's field of view was only a few degrees. During commissioning of the pointing system, the gondola was often making wide, back-and-forth swings.

*p. 14, l. 8: Do you see any sign of vibration in the accelerometers?*

We have studied the data from the accelerometers, but they have not been especially revealing. While the initial release of the balloon is evident, the remainder of the data show only a constant level of background noise. No gondola rotations or coherent vibrations (e.g., from the electrical motors) are evident, though the measurement cadence of the accelerometer was only 20 Hz. Also, we suspect that the chip's location (on a circuit board within a stack of boards) resulted in its being mechanically isolated from many vibrations.

*p. 14, l. 10:  I am sorry, but I do not see any pattern at 72 s.  Please explain.*

We acknowledge that this feature, as shown in Figure 7, is relatively subtle, and we apologize for making only a passing reference to it.  We have added a new figure (with explanatory text) that focuses on the wind-velocity measurements from just a 6-second period (that includes $t = 72$ s).

*p. 14, l. 19-22:  In order to catch the GRIPS floating altitude, a further density decrease by a factor of ∼20 needs to be compensated.  Is there any information from the manufacturer whether this gain can be achieved?  Is there an automatic control of the gain or might a gain increase result in saturation in the troposphere?*

The anemometer's gains are "hard-set" by the values of resistors in its pre-amplifiers.  We recognized from early on that, for our purposes, it would be better if the gains could be digitally controlled and thus adjusted (by command and/or automatically).  This would require re-engineering the anemometer's microcontroller board.  While we did not have the time or resources for such an undertaking for TILDAE, we would very much like to attempt it for a future flight.

*p. 15, l. 1-2:  This is a very nice achievement, but flawed by the intention to measure at GRIPS floating altitude.*

Our response to the referee's "[g]eneral comments" above address this issue.

*Typos:*

*p. 6, l. 13:  double "the"*

*p. 6, l. 17:  "based" should read "base"*

*p. 8, l. 3:  "ensuring" should read "ensure"*

*p. 13, l. 16:  "directions" should read "direction"*

We thank the referee for identifying these typos.  We have corrected them.

Response to Comments from Editor J. L. Chau

*Given the reviews received and the responses of the authors.  I encourage the authors to formally upload a revised version, including an example of at least a 3-D velocity spectrum, to show qualitative the type of of unique measurements they have obtained.  I understand a parallel scientific effort is underway, but I think including a resulting 3D spectrum without discussing the scientific details or technical details on how it was obtained, would help the authors to make their current paper (and future papers) more attractive.*

We thank the editor for this suggestion.  We have modified the manuscript to include separate power-spectra for the horizontal and vertical components of the velocity field along with the spectrum of the velocity magnitude.   A set of spectra such as these provides a more complete picture of how kinetic energy is redistributed among scales in different spatial directions in an anisotropic/stratified medium.

[revised manuscript text omitted]